# Liquid Biopsy: A Family of Possible Diagnostic Tools

**DOI:** 10.3390/diagnostics11081391

**Published:** 2021-07-31

**Authors:** Battistelli Michela

**Affiliations:** Department of Biomolecular Sciences, Carlo Bo Urbino University, 61029 Urbino, Italy; michela.battistelli@uniurb.it

**Keywords:** liquid biopsies, extracellular vesicles, tumor dynamic evaluation

## Abstract

Liquid biopsies could be considered an excellent diagnostic tool, in different physiological or pathological conditions. The possibility of using liquid biopsies for non-invasive clinical purposes is quite an old idea: indeed many years ago it was already being used in the field of non-invasive prenatal tests (NIPT) for autosomal fetal aneuploidy evaluation. In 1997 Lo et al. had identified fetal DNA in maternal plasma and serum, showing that about 10–15% of cfDNA in maternal plasma is derived from the placenta, and biologic fluid represents an important and non-invasive technique to evaluate state diseases and possible therapies. Nowadays, several body fluids, such as blood, urine, saliva and other patient samples, could be used as liquid biopsy for clinical non-invasive evaluation. These fluids contain numerous and various biomarkers and could be used for the evaluation of pathological and non-pathological conditions. In this review we will analyze the different types of liquid biopsy, their potential role in clinical diagnosis and the functional involvement of extracellular vesicles in these fluids as carriers.

## 1. Introduction

Several biological body fluids, such as blood, urine, saliva, breast milk and other patient samples, can be used for clinical investigations, because they contain numerous biomarkers [1]. The biomarkers found in these liquids can be DNA, RNA, proteins or even whole cells. More recently, the use of DNA deriving from plasma or other biological fluids is emerging as an important tool with minimal invasiveness for clinical diagnosis [2]. The purpose of these new technologies’ development is to find biomarkers using liquid biopsies that present a high versatility and minimal invasiveness. Liquid biopsies can have different functions: they can be used as a diagnosis of physiological situations, inflammatory processes, and in particular they represent a valid and practical minimally invasive tool for the analysis of tumor-derived materials. This represents a useful tool for the detection of molecular biomarkers in cancer patients [3]. Liquid biopsy is currently being used as a complementary or alternative method to surgical biopsy and represents a non-invasive detection tool that overcomes recurring problems in the clinical evaluation of tumors, deriving from the lack of accessibility to tumor tissue and its clonal heterogeneity [4]. All components derived from body fluids have been shown to reflect the genetic profile of primary and metastatic lesions and provide real-time monitoring of tumor dynamics, which holds great promise for personalized medicine. The analysis of body fluids can be used not only as a tool for the evaluation of tumor markers, but has a possible role in the diagnostic evaluation of various pathologies. Within liquid biopsies we can find molecules, DNA and RNA and different types of extracellular vesicle (EV) particles which were recently discovered to be possible analytes for liquid biopsies [5,6]. Extracellular vesicles have both protective and pathological effects and can be detected in various body fluids including urine, saliva, blood, plasma, amniotic fluid, breast milk, pleural ascites, synovial fluid, and cerebral spinal fluid [7]. Predictive biomarkers aid in the selection of a personalized therapy targeting the molecular alterations within an individual’s tumor. Patient responses to targeted therapies are commonly followed by resistance to treatment. Liquid biopsy represents a promising tool for early cancer detection and could lead to significant advancement as a diagnostic tool for a variety of cancers, including melanoma, breast, colorectal, lung, liver, ovary, pancreas and stomach. Various biological fluids including peripheral blood, urine, pleural fluid, ascites, seminal fluid, and cerebrospinal fluid (CSF) are used to isolate circulating targets for diagnostic applications. However, it is difficult to preserve the biological functions and viability of analytes isolated from liquid biopsy samples, and the reliability of the targets chosen by the analyte is not yet clear [8]. Liquid biopsies could be used as an alternative to tumor biopsies to evaluate predictive biomarkers and responses to therapy, and there are many new studies evaluating the clinical effects of this new technique. Liquid biopsy analyses could become the preferred choice of evaluation for addressing the challenges of early disease monitoring, the effect of curative treatments, and early diagnosis of disease recurrence.

## 2. Components of Biological Fluids

Peripheral blood, as well as other biological fluids, can be a source of cancer material, such as circulating tumor cells (CTCs), circulating tumor DNA (ctDNA) and extracellular vesicles (Evs). These components may reflect characteristics of the current tumor stage, and therefore these “liquids”, and in particular their components, can be considered ideal tools for providing dynamic assessments of tumor genomes.

## 3. CTCs

In 1869, Ashworth, an Australian researcher, was the first to discover tumor cells in the peripheral blood by proposing the concept of CTC. They are present in the blood of many patients with solid tumors [9]. CTCs are cancer cells, which break away from the primary tumor and enter the bloodstream. They do not bind to the extracellular matrix (ECM) and survive in the bloodstream due to their resistance to anoikis. This resistance offers the opportunity to acquire CTCs through non-invasive procedures, making it a powerful tool for detecting and monitoring cancer and metastases. Metastases are the leading cause of cancer-related deaths, but the mechanisms of metastatic spread are still poorly characterized. The metastatic cascade is a biologically complex process and presents a huge difference in dynamics and kinetics in the different types of cancer. Metastasis consists of several events including cell migration, local invasion, intrusion of tumor cells into the circulation, dissemination, arrest at secondary and primary sites, extravasation at distant sites, colonization, engraftment at distant sites and finally the formation of clinically detectable metastases. The timing of when CTCs enter circulation is not very clear, that is, whether they enter early or late. Over the last few decades some scholars have come to the conclusion that CTCs are identifiable cancer cells in the bloodstream of patients while DTCs are cancerous cells that have already extravasated into a secondary organ [10,11]. Furthermore, it is not yet clear whether clusters or individual CTCs play a fundamental role. In fact, more recent studies indicate that clusters show distinct characteristics with respect to individual CTCs, including phenotype, sign of gene expression, nature and mode of spread. Hence, establishing the role and significance of CTC clusters in cancer spread is extremely interesting. More and more evidence suggests that the survival of CTCs is due to multiple factors including resistance to anoikis, epithelial and mesenchymal plasticity or stem cell-like properties, so about 2.5% of CTCs form micro-metastases and about 0.01% of CTCs progress to macro-metastases [12]. Data has suggested that CTC-clusters may have 100 times more metastatic potential than single CTCs. Non-CTCs can be selected from fluids, erythrocytes can be removed by selective lysis or filtration, and leukocytes can be removed by filtration with antibodies (CD45), which are not present on CTCs. Positive CTC selection is based on the search for the target antigen (EpCAM) that is present on the CTCs. EpCAM is not expressed by all tumor cells (in the EMT phase), so this can produce more CTCs through the use of antibodies to other markers (alone or in mixture). Additional therapeutic targets and potential drug-resistance mechanisms can be revealed by detection and characterization of individual CTCs. Numerous researchers are studying the feasibility and usefulness of CTC analysis, in particular the effectiveness of real-time therapy monitoring and the monitoring and detection of molecular targets to predict drug resistance. Numerous researchers have shown that while CTCs from liquid biopsies can be a good diagnostic method, sadly, low CTC counts delay studies with cancer patients in the early stages of the disease while promising data are found in patients with advanced disease. Hence, despite the promising results obtained so far, much of the definition, identification, isolation, diagnostic and clinical utility has yet to be explored.

The analysis of CTCs could represent an interesting alternative to tissue biopsies of tumor metastases, since they are not invasive and require only a liquid sample. Therefore, the search for CTCs can lead to a breakthrough in diagnostic procedures, providing information on the molecular characteristics and persistence of these biological fluid cells both during diagnosis and during treatment to obtain prognostic information [13].

## 4. ctDNA

The ctDNA, on the other hand, represents the tumor in its entire heterogeneity, is easily obtainable from a blood sample and is repeatable, allowing the monitoring of the state of the disease and providing a picture of the evolution of the tumor to identify the genetic changes that may occur, while identify what is responsible for relapses, progression and development of drug resistance. The ctDNA is able to offer not only information relating to the genetic profile of the primary lesion, as occurs with traditional biopsy (tissue DNA), but also of metastases. The ctDNA contained is derived from all tumor sites providing the potential needed to monitor the disease and its progression in real time. The ctDNA is detectable not only in the blood, but also in other body fluids such as urine, saliva, breast milk and synovial fluid. During tumor progression, ctDNA is released from apoptotic or necrotic tumor cells, providing information on the genomic composition of primary and secondary tumors.

Both genetic and epigenetic alterations are involved in the development of cancer and the possibility of detecting these alterations via liquid biopsy is becoming a clinical reality. The analysis of ctDNA may, in the future, be performed using these sources [14].

## 5. Extracellular Vesicles

A new possibility of cell–cell communication mediated by membranous extracellular vesicles (EVs) has been recently proposed and is the subject of numerous molecular and morpho-functional studies. Therefore, the direct contact between two or more cells and the interaction between the ligand and the receptor that, for a long time, had been considered the only mechanism of cellular communication, has been replaced by this new mechanism of exchange of information [15]. The term “EV” refers to all types of particle which are naturally released by cells, are bounded by a lipid bilayer and cannot replicate [16]. The different particles that are identified with the common term “extracellular vesicles” differ in their biogenesis, size, physical properties, molecular composition and their function within the body. Extracellular vesicles are signal-related organelles released by many cell types and highly conserved in both prokaryotes and eukaryotes. All organisms produce extracellular vesicles and both normal and dying cells release membrane-bound vesicles such as exosomes, micro-vesicles and apoptotic bodies. Recent studies have shown that extracellular vesicles can cause both protective and pathological effects, depending on the precise condition. They can be detected in body fluids including urine, saliva, blood, plasma, amniotic fluid, breast milk, pleural ascites, synovial fluid and cerebrospinal fluid [17,18].

Cells can release three different types of extracellular vesicle, which were once thought to have the sole function of eliminating waste substances. Today, after careful analyses, it has been discovered that they carry molecules of various types, playing a fundamental role in cell communication.

EVs of endosomal origin are called exosomes and most have a diameter between 40 and 120 nm, while vesicles derived from plasma membrane budding can reach a diameter of 100–500 nm and have been called micro-vesicles, ectosomes, vesicles or microparticles [19,20]. Apoptotic cells can also release a variety of IVs via apoptotic membrane blebbing, formation of membrane protrusions such as microtubule spikes, apopto-pods, and beaded apopto-pods. EVs derived from apoptotic cells are commonly referred to as apoptotic bodies and most are from 500 nm to 2 μm in diameter, although the formation of smaller vesicles has also been reported during apoptosis progression [21,22,23]. Although the mean size of various EV subtypes differs, it is currently not possible to achieve an accurate separation of EV subtypes based on size, biochemical properties or surface markers. EVs include large particles (LEVs; also known as micro-vesicles) and small particles (sEVs or exosomes), which differ in both their intracellular origin and in the load they carry [24]. EVs and sEVs appear to reflect the molecular characteristics of their cells of origin and modulate the phenotype of recipient cells both in a paracrine and systemic way [25].

These vesicles are released into body fluids and then act on target cells by releasing information (Figure 1).

## 6. Liquid Biopsies

Liquid biopsies represent a new diagnostic technique which allows the evaluation of physiological or pathological changes within the organism by exploiting the contents of body fluids. The analysis of body fluids is not very invasive, fast and inexpensive, therefore it is fundamental in various situations [26], as it is preferrable to tissue biopsy (Figure 2). Early diagnosis offers many ways to improve intervention and positive outcomes for patients with various medical conditions [27]. Those who could benefit most from early diagnosis and treatment are cancer patients, whose survival rate decreases due to diagnostic and therapeutic delays. Most patients are already in the mid to late stages by the time they are diagnosed with cancer and by that time the most appropriate period for treatment has been missed. Since mortality rate could be reduced with an early diagnosis, the study of the material transported by biological liquids was undertaken [28]. Liquid biopsy makes early screening possible by tracing peripheral free circulating DNA (cfDNA), cell free RNA, circulating proteins, and other elements. Liquid biopsies offer the opportunity to detect, analyze, and monitor cancer in various body fluids without the need to take fragments of cancerous tissue. Within these liquids various biological elements can be identified, such as circulating tumor cells (CTCs) and nucleic acids free from extracellular vesicle (Figure 3). In addition to being a non-invasive or minimally invasive procedure, it should represent a better view of tumor heterogeneity and allow for real-time monitoring of cancer evolution [29]. Recent technological and molecular advances have allowed great progress both in the possibility and capacity to purify the components of liquid biopsy and in the analysis of the latter.

In fact, primary tumor tissue happens to be inaccessible sometimes and does not always provide enough information to stratify individual patients to the most promising therapy. In addition, a reanalysis of metastatic lesions by needle biopsy is possible, but invasive, and limited by the known intra-patient heterogeneity of individual lesions. These hurdles might be overcome by analyzing tumor cells or tumor cell products in blood samples (liquid biopsy), which is the most common fluid for clinical application and might reflect in principle all subclones present at that specific time point and allow sequential monitoring of disease evolution. Liquid biopsies inform on circulating tumor cells as well as tumor-derived cell-free nucleic acids, exosomes and platelets. Here, we will introduce the different approaches to blood-based liquid biopsies and discuss the clinical applications.

In this review we will make an excursus of the main liquid biopsies that could improve diagnoses and interventions in different pathological conditions.

## 7. Blood

Blood is the most commonly used biofluid in the search for tumor biomarkers [30]. A sample can be obtained in a simple and non-invasive way, which provides dynamic information on the progress and evolution of the type of tumor. Tumor marker proteins, circulating nucleic acids, circulating tumor DNA (ctDNA) and circulating tumor RNA (ctRNA) are present in plasma or serum, which can be investigated using highly sensitive genomic techniques [31].

Blood represents a liquid biopsy that could be used to select unique biomarkers derived from patients’ tumors. A close correlation between markers and diseases has been highlighted in the literature, therefore, these markers could play a leading role in monitoring diseases and especially cancer. Circulating cancer cells (CTCs) in blood come from primary tumors or secondary metastatic sites and the assessment of CTC alone could be used to correlate prognoses. The analysis of individual circulating cells could allow the characterization the tumor’s heterogeneity and also of the drug therapy and its effects [32,33]. Inside the blood fluid, free circulating DNA (ctDNA) has a relevant importance and many authors have studied the presence of ctDNA in liquid biopsy. Several studies have suggested its potential role for early cancer detection and prediction of cancer recurrence in several cancers with a specificity higher than 99% [34]. Its low percentage requires a high sensitivity from the extraction techniques. Recently, in addition to these two markers, the presence of extracellular vesicles has been identified within the blood fluid [35]. These nano-sized vesicles, which were initially thought to be vesicles for waste management, have a critical role in the initiation, progression and metastasis of cancer. The characterization of these vesicles within blood is becoming critical for early diagnosis and intervention in many cancers.

This fluid is already used for different cancer diagnoses, in particular for melanoma, lymphoma [36], oral [37], lung [38], gastrointestinal [39], colono-rectal [40], pancreatic, liver [41], ovarian [42], prostate [43], breast [44] and for identification of pathological conditions; in particular, it is also used for evaluation of Sjogren’s syndrome, a systemic autoimmune disease characterized by dryness of the mouth and eyes [45].

## 8. Urine Fluid

Urine has been and can be used for clinical analysis, because it is a good source of biomarkers for urinary and non-urinary tract diseases [46]. Urine tests are non-invasive and in addition to being effective for assessing pregnancy, they can be used to monitor urinary tract disorders, and other diseases such as diabetes and hypertension [47,48,49]. In the last twenty years, with the discovery of extracellular vesicles and the analysis of their release and their function, it has been shown that DNA derived from the tumor and spread through circulation can be detected in urine [50].

Urine has become one of the most interesting bio-fluids in clinical practice due to its easy collection method, its availability in terms of quantities, and non-invasiveness. Furthermore, compared to blood, urine is a less complex and relatively clean biofluid, with the only relatively abundant protein being uromodulin [51]. In fact, it has recently been shown that the DNA released by the cells into the circulation can be filtered through the kidneys in urine, therefore their urine collection, which is completely non-invasive, can act as an alternative source of body fluids in order to detect cancer-related molecular markers present in the circulation [52]. This urine cfDNA can provide reliable and reproducible information on cancer-specific DNA alterations, a discovery that is potentially useful for cancer diagnostics and monitoring. Different authors have reported that human urine contains DNA derived from both urinary tract cell debris and circulation [53,54], but other authors have shown that urine contains fragmented DNA that originates from organs outside the urinary tract and enters the circulation, and urine is therefore a viable substrate for detecting ctDNA markers [55]. This fluid can be used for inflammatory disease detection and also prostate cancer evaluation [43].

## 9. Saliva

Saliva represents a very attractive sample and an advantage for bioanalysis as it represents a clinical analysis that is easy to obtain, even from infants and the elderly, is non-invasive and non-stressful and is therefore frequently used in the search for biomarkers in various pathological situations, including stress [47], inflammation of the oral cavity [56] and cancer [57]. The sampling procedure, however, requires some precautions in order to obtain accurate and reliable results. Different analytes are present in saliva in different concentrations, but they are highly correlated with serum concentrations. Regarding the different biomarkers considered, some factors can influence the analysis, so the patient should not eat, smoke, drink or brush their teeth for at least one hour before collection. In the diagnosis of oral cancer, the most commonly used and effective technique is tissue biopsy with histological evaluation, but this technique requires specific training and it is invasive, painful, time-consuming and expensive [58,59].

Clinical diagnostic technologies for early diagnosis of oral cancer are oral transepithelial brush biopsy kits such as Oral CDx^®^ brush biopsy, tolonium chloride or toluidine blue dye, salivary diagnostics, and finally optical imaging systems [60,61,62].

All of these methods have their advantages and disadvantages. Over the years there has been an increasing need for an early diagnosis of pre-cancer and cancer through the non-invasive analysis of salivary biomarkers [63,64,65,66].

Various saliva-based biomarkers have been proposed for early detection of oral pre-cancer and cancer [58].

Since late 1992 to date, more than 100 studies have been published on more than 120 saliva biomarkers (salivary constituents, proteomic, transcriptomic, genomic and metabolomic analytes) and these have been suggested as potential diagnostic tools for oral cancer and pre-cancer diagnosis [67] and also for diagnosis of Sjogren’s syndrome [45].

## 10. Breast Milk

Breast milk is a liquid containing cells involved in the immune response and various soluble proteins which have the function of participating in the maturation of the baby’s immune system. Some authors have demonstrated the presence of exosomes inside colostrum and mature breast milk, which have a fundamental content for the development of the baby’s immune system [68].

The isolated vesicles have the typical morphology and dimensions of exosomes: in fact, they have a diameter of between 30 and 100 nm and are delimited by a double limiting membrane derived from the endosome secreted by a wide range of cell types.

Breast milk can also represent a potential and valid method to detect breast cancer through the biochemical monitoring of proteins and other molecules that we can find in it, but also in other body fluids such as nipple aspiration, ductal wash, tears, urine, and saliva. Of all these fluids, breast milk provides access to a large volume of breast tissue in the form of exfoliated epithelial cells, and to the local breast environment through the release of vesicles. Therefore, breast milk testing is a non-invasive method with significant potential for BC risk assessment. Here we have analyzed human breast milk using mass spectrometry (MS)-based proteomics to build a biomarker signature for early BC diagnosis [69].

Human milk exosomes have been shown to have a regulatory effect on the immune system and may also have an antitumor impact. These vesicles could be used, thanks to their stability and versatility, as vectors for pharmacological treatments [70,71]. The crucial role of breast milk is evaluation for the presence of tumoral marker, but it has a limited role in clinical evaluation [70]. Although biomarker-based studies mostly focus on blood-derived CTCs, ctDNAs and EVs, as most cells in our body release them into the blood, breast fluids, particularly milk, may be full of biomarkers. Therefore, breast milk may be a more reliable source for finding specific biomarkers that allow for a reliable investigation for breast cancer. Breast milk also has a regulatory effect on the immune system, and this function could be fundamental in antitumoral treatment [44].

## 11. Synovial Fluid

Inside synovial fluid the presence of exosomes, small vesicles with a diameter of 30–100 nm, has also been demonstrated: exosomes perform a fundamental function in the communication between the cells of articular devices. They are vesicular containers that carry proteins, RNA and DNA molecules. Additionally, lncRNAs are present within synovial fluid, a different class of RNA transcripts more than 200 nucleotide-long which is potentially limited in protein coding. IncRNAs, which have been extensively explored in cancer [72], appear to play an important role in osteoarthritis, although there are not yet many studies that have investigated the role of exosomal lncRNAs as biomarkers in osteoarthritis. The analysis of vesicles and biomarkers could allow the distinguishing of early OA and late OA [73], therefore their analysis could become fundamental for an early intervention in this disabling pathology.

Several recent studies have analyzed synovial fluid for miRNAs, cytokines and proteins to better understand the pathophysiological status of OA.

Extracellular (EV) vesicles derived from cells, when released into the synovial fluid (SF) of inflamed joints of patients with osteoarthritis (OA) and rheumatoid arthritis (AR), play a significant role in disease progression, triggering and contributing to the spread of the inflammatory process and participating in tissue degeneration. These vesicles contain numerous autoantigens implicated in autoimmune diseases and can induce the release of proinflammatory cytokines and growth factors from synoviocytes in vitro. There are few studies of these synovial fluid vesicles, but it is now possible to study them to characterize the progression stage of OA and AR [74]. The recent bibliography demonstrates that this liquid biopsy is currently use, in particular for inflammatory or osteoarthritis detection, but it is possible that this fluid has a crucial role in cancer evaluation (Table 1).

## 12. Conclusions

The concept of liquid biopsy was introduced to oncology with the potential to revolutionize the management of cancer patients, eliminating the invasive procedures needed to obtain tissue samples, and to provide information on therapy response and disease relapse.

In recent years, biological fluids analysis, represents a new particularly important technique for the analysis of tumor-derived material in the blood and other body fluids of cancer patients. The components of LB are mainly CTC, circulating nucleic acids free of cells, exosomes, micro-vesicles and platelets. These elements, that make up the range of tumor components in the blood, are able to reflect the genetic profile of both primary and metastatic lesions and provide real-time monitoring of the tumor, holding great promise for personalized medicine. Its importance derives from the possibility of performing it in different biofluids, overcoming the limits of tissue biopsy such as the recurring problems in the clinical evaluation of tumors that derive from the lack of accessibility to the tumor tissue and its clonal heterogeneity. LB is a complementary or alternative method for tissue biopsy. This represents a minimally invasive detection tool for molecular biomarkers which, following further investigations and through the improvement of techniques, will make diagnosis and therapeutic intervention more and more effective. Although the most commonly used liquids are blood and urine, numerous studies have been published in recent years on different biological liquids.

The analysis of the components of biological fluids could be an excellent tool for early diagnosis, to follow the course of the disease and the effect of the treatments. This technique, on which we already find numerous studies, still has limitations that should be carefully evaluated for optimization of this new technique. The first problem related to tests based on the presence of circulating tumor cells concerns the relationship between their presence and the possible development of a tumor. The cells of our body undergo continuous mutations, which are often spontaneously repaired by our cellular defense systems and which do not give rise to tumor masses. Therefore, finding a cancer cell circulating in the blood of a healthy, symptom-free person does not mean that there is actually a developing tumor. On the contrary, many tumors, before giving rise to circulating cells, need to reach a fair size and aggressiveness. Furthermore, not all circulating malignant cells give rise to metastases: many, thanks to the action of the immune system, are unable to reach the target organs and die before they can do damage. Further studies are needed to validate a technique that could improve the clinic, especially in the oncology field.

## Figures and Tables

**Figure 1 diagnostics-11-01391-f001:**
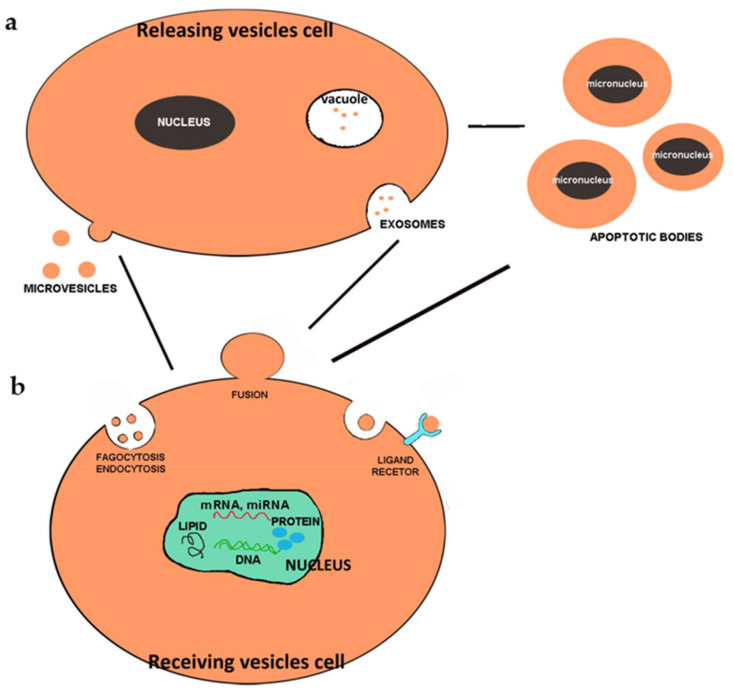
Scheme showing the various types of extracellular vesicle released by the cells (**a**) and the possible mechanisms of interaction with other cells (**b**).

**Figure 2 diagnostics-11-01391-f002:**
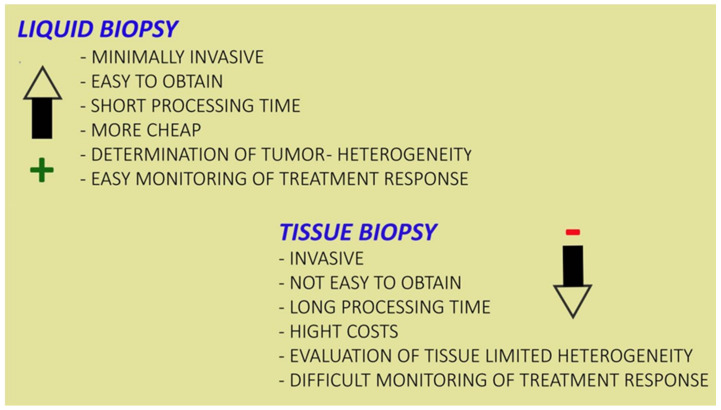
Schematic representation of advantages and disadvantages of liquid and tissue biopsies. + represent advantages of liquid biopsy clinical application, while − represent disadvantages.

**Figure 3 diagnostics-11-01391-f003:**
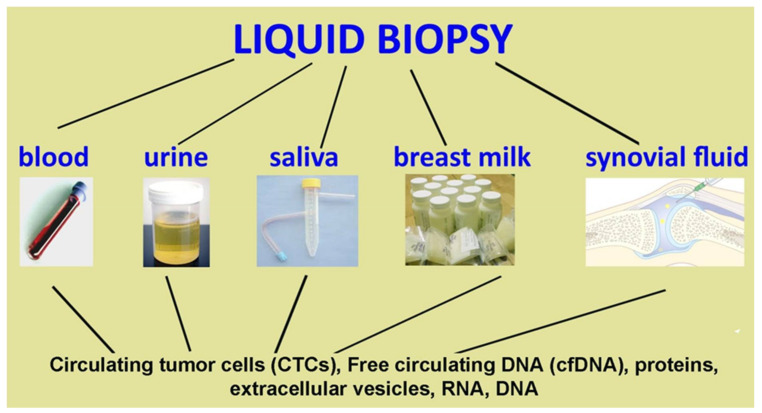
Scheme of different biologic fluids and elements contained in these fluids; in the figures it is possible to see the presence of the same elements in all body fluid.

**Table 1 diagnostics-11-01391-t001:** Summary table representing the main uses of different liquid biopsies.

Liquid Biopsy	Possible Application
Blood	Oral cancer
	Lung cancer
	Gastrointestinal cancer
	Colorectal cancer
	Pancreatic cancer
	Liver cancer
	Ovarian cancer
	Prostate cancer
	Breast cancer
	Sjogren’s syndrome
Urine	Prostate cancer
	Urologic cancer
	Colorectal cancer
Saliva	Urinary inflammatory diseases
	Oral cancer
	Sjogren’s syndrome
Breast milk	Breast cancer
Synovial fluid	Osteoarthritis
	Rheumatoid arthritis
	Joint inflammatory diseases

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
