# Peer review of "Liquid Biopsy: A Family of Possible Diagnostic Tools"

_diagnostics, 2021, doi:10.3390/diagnostics11081391_

Round 1

Reviewer 1 Report

Recommendation: Major revisions

Comments:

This manuscript describes the role of Liquid Biopsy in the field of diagnostics. The author needs to address the following comments and revise the manuscript accordingly.

  1. This manuscript is in need of substantial editing, section arrangement and style improvement.
  2. Please consider to modify the title. Current manuscript is not a concise review.
  3. Consider to include two separate sections for CTCs and cfDNA before EVs. Please consider to focus on clinical applications and include the following references:
    1. Roy, D.; Tiirikainen, M. Diagnostic Power of DNA Methylation Classifiers for Early Detection of Cancer. Trends Cancer 20206, 78–81. DOI:https://doi.org/10.1016/j.trecan.2019.12.006.
    2. Zhang X, Ju S, Wang X, Cong H. Advances in liquid biopsy using circulating tumor cells and circulating cell-free tumor DNA for detection and monitoring of breast cancer. Clin Exp Med. 2019 Aug;19(3):271-279. Epub 2019 Jun 12. PMID: 31190187. doi: 10.1007/s10238-019-00563-w
    3. Vogl TJ, Riegelbauer LJ, Oppermann E, Kostantin M, Ackermann H, Trzmiel A, et al. (2021) Early dynamic changes in circulating tumor cells and prognostic relevance following interventional radiological treatments in patients with hepatocellular carcinoma. PLoS ONE 16(2): e0246527. https://doi.org/10.1371/journal.pone.0246527.
    4. Roy, D.; Taggart, D.; Zheng, L.; Liu, D.; Li, G.; Li, M.; Zhang, K.; Etten, R.A.V. Abstract 837: Circulating cell-free DNA methylation assay: Towards early detection of multiple cancer types. Cancer Res. 2019, 79, 837.
  4. Please note that it is your responsibility to obtain permission to reproduce copyrighted material (i.e. figures, tables or excerpts that have been published online or in print) from the publishers of the original material. Consider to mention.
  5. Page2, line54: Please consider to highlight applications of EVs for early detection. Include the following recent references:
    1. Lee, K.; Fraser, K.; Ghaddar, B.; Yang, K.; Kim, E.; Balaj, L.; Chiocca, E.A.; Breakefield, X.O.; Lee, H.; Weissleder, R. Multiplexed Profiling of Single Extracellular Vesicles. ACS Nano 2018, 12, 494–503.
    2. Roy, D.; Pascher, A.; Juratli, M.A.; Sporn, J.C. The Potential of Aptamer-Mediated Liquid Biopsy for Early Detection of Cancer. Int. J. Mol. Sci. 2021, 22, 5601. https://doi.org/10.3390/ijms22115601
  6. Page3, line 95: A section for Liquid Biopsy at this part needs to be rearranged.

Author Response

I would like to thank the reviewer for ideas and suggestions that allowed me to improve my paper. I have revised the paper as suggested by reviewers.

This manuscript is in need of substantial editing, section arrangement and style improvement.

I have modified  the paper following the comments and suggestions of the reviewer.

    Please consider to modify the title. Current manuscript is not a concise review.

I have modified the title of review.

    Consider to include two separate sections for CTCs and cfDNA before EVs. Please consider to focus on clinical applications and include the following references:
        Roy, D.; Tiirikainen, M. Diagnostic Power of DNA Methylation Classifiers for Early Detection of Cancer. Trends Cancer 2020, 6, 78–81. DOI:https://doi.org/10.1016/j.trecan.2019.12.006.
        Zhang X, Ju S, Wang X, Cong H. Advances in liquid biopsy using circulating tumor cells and circulating cell-free tumor DNA for detection and monitoring of breast cancer. Clin Exp Med. 2019 Aug;19(3):271-279. Epub 2019 Jun 12. PMID: 31190187. doi: 10.1007/s10238-019-00563-w
        Vogl TJ, Riegelbauer LJ, Oppermann E, Kostantin M, Ackermann H, Trzmiel A, et al. (2021) Early dynamic changes in circulating tumor cells and prognostic relevance following interventional radiological treatments in patients with hepatocellular carcinoma. PLoS ONE 16(2): e0246527. https://doi.org/10.1371/journal.pone.0246527.

Roy, D.; Taggart, D.; Zheng, L.; Liu, D.; Li, G.; Li, M.; Zhang, K.; Etten, R.A.V. Abstract 837: Circulating cell-free DNA methylation assay: Towards early detection of multiple cancer types. Cancer Res. 2019, 79, 837.

I have inserted this bibliography and modified the text.

    Please note that it is your responsibility to obtain permission to reproduce copyrighted material (i.e. figures, tables or excerpts that have been published online or in print) from the publishers of the original material. Consider to mention.

I have produced this figures.

    Page2, line54: Please consider to highlight applications of EVs for early detection. Include the following recent references:
        Lee, K.; Fraser, K.; Ghaddar, B.; Yang, K.; Kim, E.; Balaj, L.; Chiocca, E.A.; Breakefield, X.O.; Lee, H.; Weissleder, R. Multiplexed Profiling of Single Extracellular Vesicles. ACS Nano 2018, 12, 494–503.
        Roy, D.; Pascher, A.; Juratli, M.A.; Sporn, J.C. The Potential of Aptamer-Mediated Liquid Biopsy for Early Detection of Cancer. Int. J. Mol. Sci. 2021, 22, 5601. https://doi.org/10.3390/ijms22115601

I have inserted this bibliography.

    Page3, line 95: A section for Liquid Biopsy at this part needs to be rearranged.

I have modified the Liquid biopsy section as suggested by the reviewer.

Reviewer 2 Report

Major:
1. Figures should be improved both in terms of representation and scientific details.

2. At the moment, the article is a bit vague; not many real examples are discussed in the subsections e.g., in blood, the author should describe some examples of cancers where previous publications attempted to diagnose early cancer. Similarly, in other sections.

3. It is known that liquid biopsies, e.g., detecting cancers from the blood via cfDNA do not produce accurate results. Many false-negative results indeed. That is why have not been approved these diagnostic measures yet. The author should discuss in detail the limitations and shortcomings of liquid biopsies. The authors should also discuss how this scenario could be improved to make liquid biopsy the diagnosis of choice.

4. A table summarizing different types of liquid biopsies and their application, along with published studies, is highly desirable.

  Minor:
5. Give examples of how EVs have been or could be used in the diagnosis or other clinicopathological interventions.

6. typo in abstract- lines 14, 15.

7. Sentence structure in many parts is not proper e.g. 55-60. I would recommend the author to get the article edited for language by an English native speaker.

Author Response

I would like to thank the reviewer for ideas and suggestions that allowed me to improve my paper. I have revised the paper as suggested by reviewers.

Reviewer 1:

    Figures should be improved both in terms of representation and scientific details.

I have modified the figure and the legend figures as suggested by the reviewer.

2. At the moment, the article is a bit vague; not many real examples are discussed in the subsections e.g., in blood, the author should describe some examples of cancers where previous publications attempted to diagnose early cancer. Similarly, in other sections.
I have modified: introduction and conclusion and also other part that you can fine in red.

    It is known that liquid biopsies, e.g., detecting cancers from the blood via cfDNA do not produce accurate results. Many false-negative results indeed. That is why have not been approved these diagnostic measures yet. The author should discuss in detail the limitations and shortcomings of liquid biopsies. The authors should also discuss how this scenario could be improved to make liquid biopsy the diagnosis of choice.

In the conclusion I have inserted the limitations of this tecniques.

4. A table summarizing different types of liquid biopsies and their application, along with published studies, is highly desirable.

I have enclosed the application example and summarizing table.

Minor:
5. Give examples of how EVs have been or could be used in the diagnosis or other clinicopathological interventions.

I have enclosed the example

6. typo in abstract- lines 14, 15.
I have modified the text.
7. Sentence structure in many parts is not proper e.g. 55-60. I would recommend the author to get the article edited for language by an English native speaker.

The paper was reviewed by an English speaking expert, I included her name in the acknowledgement.

Round 2

Reviewer 1 Report

This version of the manuscript is improved. Please publish.

Reviewer 2 Report

The authors addressed my points satisfactorily and the manuscript can now be considered for publication!